# Adaptive Differential Privacy for Language Modeling

**Xinwei Wu[1][†], Li Gong[2], Deyi Xiong[1]** [*]
[1]College of Intelligence and Computing, Tianjin University, Tianjin, China
[2]ByteDance Lark AI, Beijing, China
`{wuxw2021,dyxiong}@tju.edu.cn, franck.gong@bytedance.com`

## Abstract

Although differential privacy (DP) can protect language models from leaking privacy, its indiscriminative protection on all data points reduces its practical utility. Previous works improve DP training by discriminating private and non-private data. But these works rely on datasets with prior privacy information, which is not available in real-world scenarios. In this paper, we propose an **Adaptive Differential Privacy** (ADP) framework for language modeling without resorting to prior privacy information. We estimate the probability that a linguistic item contains privacy based on a language model. We further propose a new Adam algorithm that adjusts the degree of differential privacy noise injected to the language model according to the estimated privacy probabilities. Experiments demonstrate that our ADP improves differentially private language modeling to achieve good protection from canary attackers.

## 1 Introduction

Language modeling is a foundation problem in natural language processing (Bommasani et al., 2021). Recent large language models (Brown et al., 2020; Zeng et al., 2021) are usually trained at scale. Unfortunately, large language models have a tendency to remember training data in the absence of appropriate privacy protection mechanisms (Carlini et al., 2019, 2021). Since data, which are usually collected from public sources, e.g., tweets, blogs, may contain sensitive information (personal address, SSN numbers, and so on) learning a safe large language model has become increasingly important.

In recent years, differential privacy (Dwork, 2008; Dwork et al., 2014) has become a key privacy preservation method, which attempts to ran-

domize the training algorithm so that the model does not rely too much on any single training instances. Abadi et al. (2016) propose Differential Private Stochastic Gradient Descent (DP-SGD) to protect deep learning models by adding random noise to gradients. However, traditional differential privacy ignores individual attributes of data (McMahan et al., 2018). This overly pessimistic privacy protection results in poor performance or even mis-convergence of training for differentially private language models (Anil et al., 2021). Therefore, approaches are proposed to mitigate this problem by treating private and non-private data separately during the DP training process, such as selective differential privacy (Shi et al., 2021) and sensory-based privacy-$\chi$ (Qu et al., 2021). These methods require training data to provide privacy information as a hard label. Unfortunately, it is usually difficult and expensive to manually annotate privacy labels to data. Other studies (Xu et al., 2019; Tesfay et al., 2019) learn to detect privacy information in unstructured texts. However, the prerequisite is knowing keywords or reference texts of privacy information (Neerbek, 2020). Therefore, learning differentially private language models on data without prior privacy information is an open problem yet to be investigated.

In this paper, we propose an **Adaptive Differential Privacy** (ADP) framework without resorting to prior privacy information. The basic assumption behind ADP is that linguistic items containing private information do not occur frequently in real-world texts. Hence, the probability that a linguistic item contains privacy information (hereinafter *privacy probability*) is inversely proportional to the frequency of the linguistic item occurring in the dataset. With this assumption, we can estimate the privacy probability of a linguistic item based on a language model. After estimating these probabilities, we relax the constraint of differential privacy, and propose an adaptive differential privacy

---

[*]Corresponding author.
[†]Work done while this author was an intern at BtyeDance Lark AI.

method, which adjusts the Guassian noise of differential privacy based on privacy probabilities. To enable this adaptive differential privacy strategy, we further present Adaptive-DP-Adam Algorithm to train differentially private language models.

To evaluate our approach, we train transformer-based language models, and compare the performance of adaptive differential privacy against traditional differential privacy methods. Additionally, we verify the protection effectiveness of ADP models with canary attackers (Carlini et al., 2019). The results suggest that our adaptive differential privacy method can achieve good performance and protection from canary attackers.

The main contributions of this paper are three-fold.

- We propose a method to automatically estimate the probability that a linguistic item contains privacy information, relaxing the requirement of prior privacy information of previous methods.

- A new Adaptive-DP-Adam algorithm is proposed, which adaptively adjusts the magnitude of differential privacy noise to be injected into language models according to privacy probabilities.[1]

- We conduct experiments to validate the effectiveness of the proposed adaptive differential privacy in improving the performance of differentially private models and protecting sensitive information.

## 2   Related Work

Large language models (Brown et al., 2020; Zhang et al., 2020) have been attracting growing attention. Powerful large language models can achieve substantial improvements on a wide range of downstream NLP tasks. Unfortunately, large language models have a tendency to memorize training data (Carlini et al., 2019). Carlini et al. (2021) have successfully induced GPT-2 (Radford et al., 2019) to output sensitive information in its training data.

Differential privacy (Dwork, 2008; Dwork et al., 2014) is widely used to protect private information of data. Abadi et al. (2016) propose the DP-SGD algorithm to train deep learning models, and apply moment accounting to calculate cumulative privacy loss during training. Although DP-SGD can

---

[1]Code is available at https://github.com/flamewei123/ADP.

limit the risk of leaking information from training data, random noise on gradients usually degrades corresponding models (Li et al., 2021), and even cause training to not converge when a large model is trained.

To improve DP-SGD, one way is to change training settings (Li et al., 2021; Hoory et al., 2021), e.g., increasing the batch size or decreasing clipping norm. However, these methods are usually at a higher cost. Other attempts to improve the utilization of dataset information by relaxing the constraints of differential privacy. For example, Ebadi et al. (2015) propose personalized differentiated privacy to provide different levels of privacy protection for different users. Kotsogiannis et al. (2020) develop one-sided differential privacy that only protects sensitive users. Shi et al. (2021) introduce Selective Differential Privacy to add noise only into private data. These methods all need to know which items in the dataset contain private information, which is prohibitively expensive for large-scale datasets. There are some previous works (Xu et al., 2019; Tesfay et al., 2019) detecting sensitive information in unstructured texts, but relying on labeled keywords or reference texts.

## 3   Preliminary

We will introduce differential privacy (Dwork, 2008; Dwork et al., 2014), and the DP-SGD algorithm (Abadi et al., 2016) as preliminaries in this section.

### 3.1   Differential Privacy

Intuitively, an algorithm is $(\epsilon; \delta)$-DP if the output of the algorithm cannot be used to probabilistically determine the presence of a single record in the dataset by a factor of $e^\epsilon$. Formally, an algorithm $A$ satisfies $(\epsilon; \delta)$-DP if for all datasets $(D_1; D_2)$ that differ from each other by at least one instance, and for any set $S$, we have $P\{\mathcal{A}(\mathcal{D}_1) \in S\} \leq e^\epsilon P\{\mathcal{A}(\mathcal{D}_2) \in S\} + \delta$, where smaller $\epsilon$ values indicate a stronger privacy protection.

### 3.2   DP-SGD Optimization

The basic idea of DP-SGD is to clip each example gradients and add noise during model training.

Specifically, for a batch of size $L$, the loss function is $\mathcal{L}(\theta) = \frac{1}{L}\sum_{x_i} \mathcal{L}(x_i; \theta)$. For each sample $x_i$ in the batch, the gradient of $g(x_i)$ is first cut using the $l_2$ norm according to the gradient clipping level $C$, so that the maximum value of loss does

not exceed $C$:

$$g(x_i) = \frac{1}{\max\{1, \|\nabla_\theta \mathcal{L}(x_i; \theta)\|_2 / C\}} \nabla_\theta \mathcal{L}(x_i; \theta).$$

(1)

For a batch $L_t$, after the sum of clipping gradients of all samples in $L_t$ is calculated, the Gaussian noise $z \sim \mathcal{N}(0, \sigma^2 C^2 I)$ is added to the sum of gradients. Hence a new gradient $\tilde{g_{L_t}}$ required for back propagation is computed as follows:

$$\tilde{g_{L_t}} = \frac{1}{L} (\sum_{x_i} g(x_i) + z_t).$$

(2)

The smaller $C$ can lead to more stable training. And a smaller value of $\sigma$ indicates smaller noise $z$.

## 4 Adaptive Differential Privacy

In this section, we will elaborate the proposed **Adaptive Differential Privacy**. First, we introduce a method to evaluate the privacy probability of a linguistic item. Second, we propose an adaptive noise method, which adjusts the noise magnitude according to the privacy probability of an item in DP-SGD process. Finally, an Adam gradient optimization algorithm based on adaptive noise is proposed.

### 4.1 Privacy Probability Evaluation

The range of privacy is not fixed but relying on its owner, which makes it hard to judge the privacy. To solve this problem, we introduce the following assumption.

**Assumption 1**: *Texts containing privacy information do not occur frequently in a large dataset.*

We assume that the probability of texts containing private information is related to the frequency of texts appearing in dataset. Hence, the judgment of privacy can be transformed into the evaluation of the text frequency, which means the privacy probability of a token sequence is in direct proportion to the frequency of this sequence.

We then introduce a simple yet effective method to measure the frequency of text based on large-scale pre-trained language models. Giving a token sequence $s = x_1, x_2, ..., x_n$, the perplexity of the sequence is computed as follows:

$$\mathcal{P}(s) = \exp(-\frac{1}{n} \sum_{i=1}^{n} \log f_\theta(x_i | x_1, ..., x_{i-1})).$$

(3)

When the perplexity is low, it indicates that the average probability of text prediction is high. Large language models like GPT use a huge amount of text data for training. Hence, we consider such a large language model to be a trustworthy estimator.

The perplexity from a trustworthy language model is inversely proportional to the occurrence frequence of the text $o(s) \propto \frac{1}{\mathcal{P}(s)}$, and the privacy probability of $s$ is proportional to the perplexity of $s$: $\rho(s) \propto \mathcal{P}(s)$. Based on this, we propose a formula for calculating the privacy probability:

$$\rho(s) = \mathbf{normalize}(\mathcal{P}(s)),$$

(4)

where $s \in D$ and **normalize** is a normalization operator that transforms values into probability values (i.e., falling between 0 and 1).

The above method that estimates the privacy probability is not precise enough, which will inevitably cause some non-private and long-tail instances to be identified as private samples. However, from the perspective of privacy protection, such a cost is still acceptable.

### 4.2 Adaptive Noise

During differential privacy training, in the batch $B = s_1, s_2, ..., s_L$ of size $L$, the privacy probability of a token sequence $s_i \in B$ is $\rho(s_i)$, and the Gaussian noise of $B$ is $z_B = \mathcal{N}(0, C^2 \sigma^2 I^2)$, where $\sigma$ is a noise multiplier, and $C$ is the clipping norm. To improve the target model performance, we introduce the **privacy weight** to change the magnitude of Gaussian noise

$$\gamma_B = \frac{\sum_i^L \rho(s_i)}{L}.$$

(5)

The **privacy weight** denotes a privacy probability averaged over batch $B$. We incorporate it to the Gaussian noise:

$$z_{B adp} = \gamma_B \cdot \mathcal{N}(0, C^2 \sigma^2 I^2).$$

(6)

Through this method, we adaptively change the noise of every batch according to its privacy weight.

### 4.3 Adaptive DP Optimization

With the adaptive noise, we further develop a privacy mechanism to train models. Abadi et al. (2016) propose DP-SGD that adds Gaussian noise to gradients and applies stochastic gradient descent (SGD) to train private deep learning models. We incorporate our proposed adaptive noise into DP-SGD.

Such adapted framework is also suitable for other optimization algorithms such as Adam. The whole procedure of Adaptive-DP-Adam is described in **Algorithm 1**.

**Algorithm 1:** Adaptive-DP-Adam

1 **Input**: dataset $D = \{x_i\}_{i=1}^N$, a large
   language model $f_{LM}$, loss function $L(\theta)$
2 **Parameters**: learning rate $\eta$, noise level $\sigma$,
   batch $B$ of size $L$, clipping norm $C$, step
   $E$, Adam parameters $\{\theta_0, m_0, m_1, \delta_1, \delta_2\}$
  1: Let $G(\varphi) = 0$
  2: **for all** $t \in T$ **do**
  3:    Sample a batch $B_t$, with sampling
      probability $L/N$
  4:    Calculate $\gamma_{B_t}$ based on Eq. (5)
  5:    **for all** $x_i \in B_t$ **do**
  6:       Clip gradients
         $\tilde{g}_t(x_i) \leftarrow g_t(x_i) \cdot min(1, C/\|g_t(x_i)\|_2)$
  7:    **end for**
  8:    Generate adaptive noise $z_t$ based on Eq. (6)
  9:    Calculate average gradients
        $\bar{g}_t(x_i) = \frac{1}{L}(z_t + \sum_{i=1}^L \tilde{g}_t(x_i))$
 10:    Update parameters $\theta$ using usual Adam
 11: **end for**
 12: **return** $\theta_T$

## 5 Experiments

### 5.1 Settings

**Dataset** We used Wikitext-103 (Merity et al., 2016) to train our model, which is a widely used dataset for language modeling from a set of verified Good and Featured articles on Wikipedia.

**Baselines** We have two baselines, one without DP (denoted by "No-DP"), and the other trained with DP-SGD (denoted by "DP-SGD"). We refer to our models trained with ADP-SGD as "ADP".

**Hyper-parameters** We used a 12-layer transformer decoder to train the language model with hidden size of 1024 and batch size of 4096, training 20 epoches with inital learning rate of $5 \times 10^{-5}$. The clipping norm $C$ was set to 0.001, and the noise multiplier $\sigma$ was 1 or 5.

### 5.2 Canary Attacker

Canary insertion is proposed by Carlini et al. (2019), which inserts random sequences called canaries into the training dataset and calculates the exposure for the inserted canaries during testing to measure whether the model memorizes these canaries. In our setting, we injected "My ID is 955320" into the Wikitext-103 dataset for 10, 100, and 1000 times to make the differences between

| model | test loss | test PPL | sigma | epsilon |
|-------|-----------|----------|-------|---------|
| No-DP | 7.08 | 256.66 | - | - |
| DP-SGD | 13.08 | 7582.65 | 1.0 | 4.22 |
| ADP | **12.65** | **4426.05** | 1.0 | 6.35 |
| No-DP | 7.08 | 256.66 | - | - |
| DP-SGD | 17.65 | 20815.23 | 5.0 | 0.1 |
| ADP | **14.85** | **8635.66** | 5.0 | 2.47 |

Table 1: The performance of language models trained by our method and baselines. We compare results by varying the noise level $\sigma$.

models more salient. Given a canary $s[r]$, and a model with parameters $\theta$, the exposure of $s[r]$ is calculated as:

$$\mathbf{exposure} = \log_2 |\mathcal{R}| - \log_2 \mathbf{rank}_\theta(s[r]), \quad (7)$$

where $\mathcal{R}$ is the set of all possible results, and $\mathbf{rank}(s[r])$ is the position of $s[r]$ in $\mathcal{R}$. The lower the exposure, the safer the model is.

### 5.3 Results

**Model Performance** We first evaluated models trained by different privacy settings on language modeling task. Both models were trained using a transformer decoder architecture. As shown in Table 1, DP-SGD performs poorly, and larger noise $\sigma$ further worses the model. In contrast, our ADP helps model to alleviate the decaying performance, and the utility grows when the noise multiplier $\sigma$ is large. Although the privacy guarantee $\epsilon$ of ADP increases compared to DP-SGD when the noise multiplier $\sigma$ is 1 and 5, the privacy guarantee of ADP is within the acceptable range. It suggests that our ADP can improve the performance of differentially private language models with tight privacy guarantee.

**Protection Against Attacker** Our second group of experiments, described in section 5.2, is to test the model memorization of private information. We evaluated models trained on the Wikitext-103 dataset injected canaries. We used text generation to evaluate the exposure of canaries from different language models. As can be seen from Figure 1, even when private item appears as many as 1000 times in the data, the ADP model performs significantly better than the non-DP model. However, exposures of the ADP model are larger than the DP-SGD model. It suggests that ADP method can protect privacy information from leaking from training data, but the protection performance is slightly worse than DP-SGD.

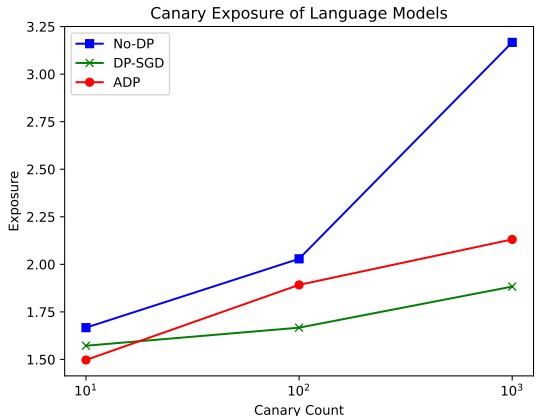

Figure 1: The exposure of canaries from different language models. All models were trained for 20 epoches.

## 6 Conclusion

We have presented a new method to estimate the privacy probability of a linguistic item when the privacy information of the dataset is not known. With estimated privacy probabilities, we propose adaptive differential privacy (ADP), to improve the model utility. We also present a privacy optimization algorithm, Adaptive-DP-Adam, to train differentially private models. Our experiments show that models trained with ADP achieve better utilities than traditional DP and are capable of protecting sensitive information from being leaked.

## Acknowledgements

The work was partially supported by a ByteDance Research Collaboration Project (PJ20210625900030) and the Natural Science Foundation of Tianjin (Grant No. 19JCZDJC31400). We would like to thank the anonymous reviewers for their insightful comments.

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
