# OpenReview forum: "Adaptive Differential Privacy for Language Model Training"
_aclweb.org/ACL/2022/Workshop/FL4NLP — FL4NLP@ACL2022_

### Official Review · Reviewer_Zgb7 · 2022-03-23
**Interesting topic and idea, but contribution is not clear**

**Rating:** 5
**Confidence:** 4

**Review:**

Overall, I think the topic of paper would be interesting for the workshop, but contribution of the paper is not clear to me. I am concerned whether there is a contribution in the method as it does not compare its proposed method with standard methods for detecting private data. I am also concerned about its empirical results as it only compares with a method published in 2016.


Below are some comments that I hope authors will find useful.

Paper describes its motivation and contribution clearly. But the following statement was not convincing for me: "There are some previous works detecting sensitive information in un-structured texts, but relying on labeled keywords or reference texts."

Instead of using the above literature, authors claim that probability of data being private is inversely related to its occurrence frequency. While this might be a good rule of thumb, authors have not verified it. Authors mention this assumption to be one of their contributions "we propose a method to automatically estimate the probability that a linguistic item contains privacy information...". Why is this a contribution? Why is this proposed method better than those methods that use keywords and reference text? Does this estimation method based on frequency of the words perform better?

Authors give social security number as one example of private data. Does their method of inferring privacy based on occurrence frequency work well in detecting SSN? Would the method gather each and every number that appears once in the text? Why should we not use known references and keywords and adopt this proposed method? Is it because this method is faster or more accurate? Authors have not compared their method with standard methods for detecting private data.

Authors use the term "non-privacy data" over and over. perhaps they mean to say "non-private data"?

There is considerable repetition in Introduction and Related Work sections.

Contribution of Abadi et al (2016) is described many times in the paper.

Algorithm 1 is already in the literature except that one line of it is modified based on equation (5). It is not clear why it has to be provided in the main text of the paper.

Method is compared with Abadi's method published in 2016. There are more recent methods in the literature that authors can consider comparing with.  Results do not appear to be convincing.

---

### Official Review · Reviewer_c46X · 2022-03-24
**This paper scales the noise to be inversely proportional to sequence frequencies to reduce exposures of privacy data. This is an interesting idea, but part of the experiments needs more explanation and can be improved.**

**Rating:** 6
**Confidence:** 3

**Review:**

Thanks for the submission! This paper proposes to treat privacy and non-privacy data differently to improve differentially private language modeling. More specifically, this paper first made a reasonable assumption that texts with privacy information do not occur frequently in a large dataset. Based on that, the algorithm estimates the privacy probability to be proportional to the perplexity of a specific sequence. Then, the authors introduce a privacy weight, representing the privacy probability, to scale the gaussian noise. Experiments show that the proposed algorithm achieved lower test loss/perplexity and at the same time reduced the exposure of canaries.

1. Since the algorithm scales the noise based on the inverse of sequence frequency, I assume this means infrequent non-privacy sequences would also end up with high perplexity? Will this be a problem?
2. Having privacy data like “My ID is 955320” occurring 1000 times seems rare. Are there any statistics or related work describing what are the reasonable configurations for evaluations?
3. I am curious about how the proposed algorithm compares to a slightly modified version which simply drops sequences below a certain frequency and then applies the same privacy weight to the rest of the dataset.
4. DP-SGD seems achieved a lower exposure compared to the proposed algorithm when the canary count is 10. What is the reason for this? This seems to be a legit problem as it's likely that privacy data will occur less than 10 times, no?

---

### Official Review · Reviewer_JsH6 · 2022-03-24

**Rating:** 4
**Confidence:** 3

**Review:**

This paper proposes a novel look at differentially private training by focusing on adding more noise to data points that are rare and less noise to popular datapoints, e.g. making an adaptive noise.

This idea is interesting, however, it contradicts multiple important assumptions of differential privacy:
1. DP says that removal of any point should result in roughly the same model, however from the proposed method not all the points are equal, e.g. some datapoints experience more noise than others. Therefore the proposed algorithm will not be differentially private.
2. Noise is correlated against a pre-trained model assuming that the training dataset is also correlated with the dataset that the pre-trained model was trained on. For example, let's assume that the dataset that I want to train my model on contains password recovery questions and answers from the users -- any common phrase (from the pre-trained dataset) will then receive less noise, but since all of those pass answers are sensitive then it will violate privacy of these secret answers. This is the weird case, as you wouldn't want to train such a model but hopefully it shows that now the guarantees depend on some pre-trained model with a custom dataset.

Lastly the performance results in Table 1 demonstrate that if you increase privacy budget epsilon then the results improve which is trivially obvious. It might have been more correct to compare performance of your algorithm with budgets of the same value.

---

### Official Review · Reviewer_7hxZ · 2022-03-27
**Concerns about the privacy definition**

**Rating:** 4
**Confidence:** 2

**Review:**

This paper proposed adaptive differential privacy. For a DP-SGD like algorithm, when adding noise, this paper proposed to use the "frequency" to scale the noise. The "frequency" is estimated by estimating perplexity based on a trustworthy language model.

My main concern is that the privacy definition is not clear to me. Is it possible to give a formal definition for adaptive DP and its accounting,  something similar to Def 2.4 in "The Algorithmic Foundations of Differential Privacy" and Theorem 1 of "Deep Learning with Differential Privacy". Without a definition, I cannot understand how the (\epsilon, \delta) bound is computed and compared.

A relatively minor concern is that it is not clear to me how to get the trustworthy language model. If it is based on public data, why it can represent the private data?

---

### Decision · Program_Chairs · 2022-03-26

Accept